

# Comparison between pystan and numpyro in Bayesian item response theory: evaluation of agreement of estimated latent parameters and sampling performance

Mizuho Nishio[1], Eiji Ota[2], Hidetoshi Matsuo[1], Takaaki Matsunaga[1], Aki Miyazaki[1] and Takamichi Murakami[1]

[1] Department of Radiology, Kobe University Graduate School of Medicine, Kobe, Japan
[2] Futaba Numerical Technologies, Iruma, Japan

Corresponding author
Mizuho Nishio,
nmizuho@med.kobe-u.ac.jp

## ABSTRACT

**Purpose**. The purpose of this study is to compare two libraries dedicated to the Markov chain Monte Carlo method: pystan and numpyro. In the comparison, we mainly focused on the agreement of estimated latent parameters and the performance of sampling using the Markov chain Monte Carlo method in Bayesian item response theory (IRT).

**Materials and methods**. Bayesian 1PL-IRT and 2PL-IRT were implemented with pystan and numpyro. Then, the Bayesian 1PL-IRT and 2PL-IRT were applied to two types of medical data obtained from a published article. The same prior distributions of latent parameters were used in both pystan and numpyro. Estimation results of latent parameters of 1PL-IRT and 2PL-IRT were compared between pystan and numpyro. Additionally, the computational cost of the Markov chain Monte Carlo method was compared between the two libraries. To evaluate the computational cost of IRT models, simulation data were generated from the medical data and numpyro.

**Results**. For all the combinations of IRT types (1PL-IRT or 2PL-IRT) and medical data types, the mean and standard deviation of the estimated latent parameters were in good agreement between pystan and numpyro. In most cases, the sampling time using the Markov chain Monte Carlo method was shorter in numpyro than that in pystan. When the large-sized simulation data were used, numpyro with a graphics processing unit was useful for reducing the sampling time.

**Conclusion**. Numpyro and pystan were useful for applying the Bayesian 1PL-IRT and 2PL-IRT. Our results show that the two libraries yielded similar estimation result and that regarding to sampling time, the fastest libraries differed based on the dataset size.

## INTRODUCTION

Item response theory (IRT) is a statistical framework used for analyzing test results and evaluating test items and test takers quantitatively. While IRT is commonly used in

educational and psychological research (*Embretson & Reise, 2000*; *Hays, Morales & Reise, 2000*; *Cappelleri, Jason Lundy & Hays, 2014*), there are several applications of IRT to medical research. For example, *Choi et al. (2010)* used IRT for constructing the computer adaptive testing system of short-form patient-reported outcome measures with the data from the Patient-Reported Outcomes Measurement Information System project. *Gershon et al. (2012)* used IRT to build the quality of life item banks for adults with neurological disorders. The most notable example of computer adaptive testing system using IRT is the National Institutes of Health Patient-Reported Outcomes Measurement Information System (PROMIS) (https://www.healthmeasures.net/explore-measurement-systems/promis). PROMIS is an NIH-funded initiative to develop and validate patient reported outcomes for clinical research and practice.

Generally, IRT is applied to the results of binary responses to the test items (*e.g.*, correct and incorrect answers). In medical diagnosis, the results of various diagnostic procedures are frequently defined as binary responses. Therefore, it is possible to apply IRT to the data of medical diagnosis. To apply IRT to the data of medical diagnosis, the following correspondence is assumed: (i) the patient as the test item, (ii) the doctor as the test taker, and (iii) the results of the binary responses obtained through medical diagnosis as test results. For example, *Nishio et al. (2020)* used IRT for analyzing the results of medical diagnoses by radiologists.

The Bayesian IRT can be implemented using probabilistic programming languages or dedicated libraries (*e.g.*, JAGS, Stan, pystan, and numpyro) (*Python Software Foundation, 2023*; *Depaoli, Clifton & Cobb, 2016*; *Carpenter et al., 2017*; *Phan, Pradhan & Jankowiak, 2019*). For example, previous studies used Stan for the implementation of the Bayesian IRT, graded response model, and nominal response model (*Luo & Jiao, 2017*; *Nishio et al., 2020*; *Nishio et al., 2022*). The recent advances in hardware and software make it possible to use the Bayesian IRT efficiently. However, there is no study comparing the efficiency of the Bayesian IRT from the viewpoint of computational cost.

The purpose of the current study was to compare the results of the Bayesian IRT implemented with two dedicated libraries (pystan and numpyro). In the current study, the Bayesian 1PL-IRT and 2PL-IRT implemented with pystan and numpyro were applied to the two types of medical data obtained from *Nishio et al. (2020)*. Our main contributions in this article are as follows; (i) the two IRT models could be fitted with pystan and numpyro, (ii) the two libraries yielded similar estimation results for the combinations of the two IRT models and the two types of medical data, and (iii) depending on the dataset size, we evaluated which package had better Markov chain Monte Carlo sampling performance. For reproducibility, our implementation of the Bayesian IRT in pystan and numpyro used in the current study is disclosed as open source through GitHub (https://github.com/jurader/irt_pystan_numpyro). Selections of this article were previously published as a preprint (*Nishio et al., 2023*).

## MATERIALS & METHODS

Because this study used the medical data obtained from *Nishio et al. (2020)*, institutional review board approval or informed consent of patients was not necessary.

**Table 1  Characteristics of two types of medical data.**

| Name of data | Number of test takers | Number of test items | Total number of binary responses |
|---|---|---|---|
| BONE data | 7 | 60 | 420 |
| BRAIN data | 14 | 42 | 588 |

## Medical data

The two types of medical data (BONE and BRAIN data) were obtained from *Nishio et al. (2020)*. Table 1 shows the characteristics of the two types of medical data. The BONE data include binary responses from 60 patients (test items) and seven radiologists (test takers), and the BRAIN data include those from 42 patients and 14 radiologists. The total numbers of the binary responses were 420 and 588 in the BONE and BRAIN data, respectively. While the data from two modalities (computed tomography and temporal subtraction) were used in *Nishio et al. (2020)*, those from one modality (computed tomography) were used in this study. As a result, the total numbers of the binary responses were half in this study, compared with *Nishio et al. (2020)*.

## 1PL-IRT

IRT is a statistical model for analyzing the results of binary responses. While there are several types of IRT models (*Gelman et al., 2013*), 1PL-IRT and 2PL-IRT were used. In the current study, latent parameters of IRT are estimated based on the results of medical diagnoses by test takers.

In 1PL-IRT, one latent parameter ($\beta_i$) is used to represent the difficulty of test item $i$, and another latent parameter ($\theta_j$) is used to represent the ability of test taker $j$. The following equations represent 1PL-IRT.

$$\Pr(r_{ij} = 1) = \frac{1}{1 + \exp(-z_{ij})}$$

$$z_{ij} = \theta_j - \beta_i$$

Here,

- $\Pr(r_{ij} = 1)$ represents the probability that the response of test taker $j$ to test item $i$ is correct,
- $\beta_i$ is the difficulty parameter of test item $i$,
- $\theta_j$ is the ability parameter of test taker $j$.

## 2PL-IRT

In 2PL-IRT, two latent parameters ($\alpha_i$ and $\beta_i$) are used to represent test item $i$. The following equations represent 2PL-IRT.

$$\Pr(r_{ij} = 1) = \frac{1}{1 + \exp(-z_{ij})}$$

$$z_{ij} = \alpha_i(\theta_j - \beta_i).$$

Here,

- $\alpha_i$ and $\beta_i$ are the discrimination and difficulty parameters of test item $i$.

## Experiments

We mainly used Google Colaboratory to run the experiments. The following software packages were used on Google Colaboratory: pystan, version 3.3.0; jax, version 0.4.4; jaxlib, version 0.4.4+cuda11.cudnn82; numpyro, version 0.10.1. Two cores of Intel(R) Xeon(R) (2.20 GHz) and NVIDIA(R) Tesla T4(R) were used as CPU and a graphics processing unit (GPU), respectively.

### Experiments for agreement of latent parameters

The Bayesian 1PL-IRT and 2PL-IRT, implemented with pystan and numpyro, were applied to the BONE and BRAIN data. For 1PL-IRT, the following prior distributions were used:

- $\beta_i \sim N(0, 2)$,
- $\theta_j \sim N(0, 2)$,

where $N$ represents a normal distribution in which the first and second arguments are the average and variance of normal distribution, respectively. For 2PL-IRT, the following prior distribution was used in addition to those of 1PL-IRT:

- $log(\alpha_i) \sim N(0.5, 1)$.

The same prior distributions of the latent parameters were used in both pystan and numpyro.

The following parameters were used for sampling using Markov chain Monte Carlo method in both pytan and numpyro: number of chains (num_chains) = 6, number of samples per one chain (num_samples) = 8,000, number of samples for warmup (num_warmup) = 2,000. For numpyro GPU version, chain_method = 'parallel' was used. After the sampling, the posterior distributions of the latent parameters were obtained for the Bayesian 1PL-IRT and 2PL-IRT. The posterior distributions of the latent parameters were then compared between pystan and numpyro.

### Experiments for computational cost

To evaluate the computation cost of 1PL-IRT and 2PL-IRT implemented with pystan and numpyro, simulation data were generated from the medical data (BONE and BRAIN data) and numpyro. The computer simulation was performed in the following steps: (i) estimating the posterior distributions of the latent parameters of 1PL-IRT and 2PL-IRT for the two types of medical data, and (ii) generating binary responses from the IRT equations and the estimated posterior distributions for the two types of medical data. The total number of binary responses in the simulation data were as follows: 420, 840, 2,100, 4,200, 8,400, 21,000, 42,000, 84,000, 210,000, and 420,000 for the BONE data; 588, 1,176, 2,940, 5,880, 11,760, 29,400, 58,800, 117,600, 294,000, and 588,000 for the BRAIN data. This means that size of the simulation data ranged from the original size to 1,000 times. To evaluate the computational cost in pystan and numpyro, the sampling time using Markov

chain Monte Carlo method was measured. In numpyro, both CPU version and GPU version were used for the sampling. The following parameters were used for the sampling in both pytan and numpyro: number of chains (num_chains) = 2, number of samples per one chain (num_samples) = 3,000, number of samples for warmup (num_warmup) = 500. For numpyro GPU version, chain_method = 'parallel' was used. Due to the limitation of Google Colaboratory, it was not possible to evaluate the sampling time in several large-sized simulation data. Therefore, in addition to Google Colaboratory, we performed the same experiment on a local workstation with Intel Core i9-9820X CPU and Nvidia Quadro RTX 8000.

## RESULTS

### Results for agreement of latent parameters

In the current study, we focused on the ability parameters of test takers, and the estimation results of test items were omitted. Tables 2–5 present the estimation results of the ability parameters of test takers. In addition, Figs. 1 and 2 show representative scatter plots of the estimation results between pytan and numpyro, which are obtained from values of Tables 2 and 5, respectively. Tables 2–5 show the mean, standard deviation, and credible interval (highest density interval) as the estimation results of the ability parameters of test takers. Based on Tables 2–5 and Figs. 1 and 2, we found that there was good agreement between pystan and numpyro for 1PL-IRT and 2PL-IRT of the BONE and BRAIN data.

From Tables 2–5, Lin's concordance correlation coefficients (CCC) (*Lin, 1989*) of the estimated mean of the ability parameters were calculated between (a) pystan *v.s.* numpyro CPU version, (b) pystan *v.s.* numpyro GPU version, and (c) numpyro CPU version *v.s.* numpyro GPU version. The results of CCC values are summarized in Table 6. The following criteria were used to evaluate CCC (*Nishio et al., 2016*; *Kojita et al., 2021*); low CCC values (<0.900) were considered to represent poor agreement, whereas higher CCC values represented moderate (0.900–0.950), substantial (0.951–0.990), and almost perfect agreement (>0.990). As shown in Table 6, the CCC values of the estimated mean of the ability parameters indicate almost perfect agreement for 1PL-IRT and 2PL-IRT of the BONE and BRAIN data.

In addition, when the number of samples were fewer (number of samples per one chain was less than 8,000), the agreement of the ability parameters was investigated. The results are shown in Table 7.

### Results for computational cost

Figures 3–6 show the sampling time for the simulation data of the BONE and BRAIN data. When original-size simulation data were used, the sampling time was shorter in pystan than numpyro CPU version. However, in the simulation data of the BONE and BRAIN data except for the original size, the sampling time was shorter in numpyro CPU version than pystan. Moreover, when the large-sized simulation data (total number of binary responses > 30,000–50,000) were used, the sampling time was shorter in numpyro GPU version than numpyro CPU version. In addition to the experiments using Google Colaboratory, the

**Table 2   Estimation results of ability parameters of test takers in 1PL-IRT for BONE data.**

| pystan | | Estimation | | | | Diagnostics | | |
|---|---|---|---|---|---|---|---|---|
| | | mean | sd | hdi_3% | hdi_97% | ess_bulk | ess_tail | r_hat |
| | theta[0] | 2.201 | 0.501 | 1.257 | 3.133 | 40373 | 36621 | 1 |
| | theta[1] | 3.387 | 0.61 | 2.25 | 4.531 | 47056 | 35326 | 1 |
| | theta[2] | 3.105 | 0.583 | 2.055 | 4.233 | 44783 | 36038 | 1 |
| | theta[3] | 2.845 | 0.556 | 1.8 | 3.882 | 44444 | 36888 | 1 |
| | theta[4] | 1.383 | 0.452 | 0.511 | 2.205 | 38698 | 37129 | 1 |
| | theta[5] | 2.401 | 0.517 | 1.443 | 3.378 | 40061 | 36388 | 1 |
| | theta[6] | 2.2 | 0.5 | 1.273 | 3.153 | 40777 | 38464 | 1 |
| numpyro CPU | | Estimation | | | | Diagnostics | | |
| | | mean | sd | hdi_3% | hdi_97% | ess_bulk | ess_tail | r_hat |
| | theta[0] | 2.2 | 0.508 | 1.277 | 3.185 | 32921 | 34003 | 1 |
| | theta[1] | 3.391 | 0.615 | 2.26 | 4.567 | 34162 | 35373 | 1 |
| | theta[2] | 3.099 | 0.586 | 2 | 4.201 | 35639 | 34188 | 1 |
| | theta[3] | 2.845 | 0.553 | 1.818 | 3.888 | 34989 | 35124 | 1 |
| | theta[4] | 1.382 | 0.454 | 0.517 | 2.225 | 29675 | 34343 | 1 |
| | theta[5] | 2.4 | 0.519 | 1.449 | 3.392 | 33463 | 34351 | 1 |
| | theta[6] | 2.198 | 0.506 | 1.267 | 3.174 | 33284 | 35760 | 1 |
| numpyro GPU | | Estimation | | | | Diagnostics | | |
| | | mean | sd | hdi_3% | hdi_97% | ess_bulk | ess_tail | r_hat |
| | theta[0] | 2.199 | 0.506 | 1.242 | 3.134 | 32251 | 34882 | 1 |
| | theta[1] | 3.384 | 0.615 | 2.252 | 4.556 | 34198 | 36149 | 1 |
| | theta[2] | 3.1 | 0.585 | 2.015 | 4.212 | 36366 | 35577 | 1 |
| | theta[3] | 2.843 | 0.556 | 1.786 | 3.875 | 34938 | 33051 | 1 |
| | theta[4] | 1.379 | 0.451 | 0.538 | 2.236 | 30942 | 34819 | 1 |
| | theta[5] | 2.4 | 0.521 | 1.466 | 3.419 | 32772 | 34828 | 1 |
| | theta[6] | 2.199 | 0.506 | 1.262 | 3.163 | 32300 | 36210 | 1 |

**Notes.**

[a]Note: theta[i] of Table means $\theta_i$ of the equation of 1PL-IRT.

results of computational cost on the local workstation are shown in Figs. 7–10. The same trend was observed when using Google Colaboratory and the local workstation.

## DISCUSSION

The current study aimed to compare the estimation results of the ability parameter of test takers using two different libraries (pystan and numpyro) for two different types of IRT models and medical data (BONE and BRAIN data). The study found that there was good agreement between pystan and numpyro for all the combinations of the IRT models and medical data; there was almost perfect agreement between pystan and numpyro in the CCC values of the estimated mean of ability parameters. The current study also compared the sampling time for the simulation data of the BONE and BRAIN data. We found that while the sampling time was shorter in pystan than numpyro CPU version for the original-size data, it was shorter in numpyro CPU version than pystan for the simulation data except

**Table 3** Estimation results of ability parameters of test takers in 2PL-IRT for BONE data.

| pystan | | Estimation | | | | Diagnostics | | |
|---|---|---|---|---|---|---|---|---|
| | | mean | sd | hdi_3% | hdi_97% | ess_bulk | ess_tail | r_hat |
| | theta[0] | 2.062 | 0.49 | 1.128 | 2.976 | 14702 | 24025 | 1 |
| | theta[1] | 3.086 | 0.605 | 1.966 | 4.231 | 17011 | 25730 | 1 |
| | theta[2] | 2.677 | 0.539 | 1.664 | 3.687 | 16745 | 26368 | 1 |
| | theta[3] | 2.585 | 0.571 | 1.534 | 3.658 | 16199 | 25432 | 1 |
| | theta[4] | 1.252 | 0.412 | 0.482 | 2.029 | 14322 | 22764 | 1 |
| | theta[5] | 2.096 | 0.536 | 1.102 | 3.111 | 13843 | 23047 | 1 |
| | theta[6] | 2.169 | 0.511 | 1.206 | 3.125 | 15188 | 26449 | 1 |
| numpyro CPU | | Estimation | | | | Diagnostics | | |
| | | mean | sd | hdi_3% | hdi_97% | ess_bulk | ess_tail | r_hat |
| | theta[0] | 2.057 | 0.486 | 1.159 | 2.991 | 15715 | 25398 | 1 |
| | theta[1] | 3.086 | 0.607 | 1.943 | 4.196 | 17702 | 27769 | 1 |
| | theta[2] | 2.676 | 0.546 | 1.679 | 3.723 | 16704 | 26124 | 1 |
| | theta[3] | 2.581 | 0.576 | 1.515 | 3.674 | 15881 | 25012 | 1 |
| | theta[4] | 1.248 | 0.409 | 0.494 | 2.037 | 15187 | 26085 | 1 |
| | theta[5] | 2.095 | 0.533 | 1.106 | 3.094 | 15093 | 24705 | 1 |
| | theta[6] | 2.167 | 0.511 | 1.194 | 3.104 | 15545 | 27676 | 1 |
| numpyro GPU | | Estimation | | | | Diagnostics | | |
| | | mean | sd | hdi_3% | hdi_97% | ess_bulk | ess_tail | r_hat |
| | theta[0] | 2.063 | 0.487 | 1.159 | 2.988 | 14397 | 25263 | 1 |
| | theta[1] | 3.09 | 0.601 | 1.975 | 4.226 | 16891 | 27636 | 1 |
| | theta[2] | 2.68 | 0.546 | 1.658 | 3.701 | 15578 | 27217 | 1 |
| | theta[3] | 2.585 | 0.576 | 1.525 | 3.682 | 14965 | 24243 | 1 |
| | theta[4] | 1.251 | 0.411 | 0.503 | 2.046 | 14136 | 23883 | 1 |
| | theta[5] | 2.096 | 0.538 | 1.105 | 3.112 | 13277 | 23510 | 1 |
| | theta[6] | 2.171 | 0.513 | 1.237 | 3.162 | 14117 | 24946 | 1 |

**Notes.**

Note: theta[i] of Table means $\theta_i$ of the equation of 2PL-IRT.

for the original size. For the large-sized simulation data, the sampling time was shorter in numpyro GPU version than numpyro CPU version.

Our results show that there was almost perfect agreement in the ability parameters of the Bayesian IRT between pystan and numpyro. This suggests that researchers can choose either library for implementing the Bayesian IRT. While numpyro requires only Python, both Python and Stan (two different programming languages) are necessary for pystan. Many practitioners and researchers may find numpyro to be simple and straightforward.

Although we used the simulation data, our results of sampling time show that the fastest libraries differed based on the total number of binary responses. Specifically, pystan was the fastest for the original-size simulation data, while numpyro CPU version was the fastest for the small-sized and medium-sized data. For the large-sized simulation data, the sampling time was shorter in numpyro GPU version than numpyro CPU version. This implies that

**Table 4   Estimation results of ability parameters of test takers in 1PL-IRT for BRAIN data.**

| pystan | | Estimation | | | | Diagnostics | | |
|---|---|---|---|---|---|---|---|---|
| | | mean | sd | hdi_3% | hdi_97% | ess_bulk | ess_tail | r_hat |
| | theta[0] | 1.615 | 0.567 | 0.569 | 2.695 | 19440 | 27845 | 1 |
| | theta[1] | 0.929 | 0.53 | −0.044 | 1.946 | 17105 | 28114 | 1 |
| | theta[2] | 1.617 | 0.57 | 0.561 | 2.698 | 19394 | 29146 | 1 |
| | theta[3] | 0.926 | 0.527 | −0.075 | 1.902 | 17568 | 27120 | 1 |
| | theta[4] | 1.139 | 0.537 | 0.145 | 2.163 | 18614 | 28549 | 1 |
| | theta[5] | 1.144 | 0.543 | 0.117 | 2.147 | 16985 | 28225 | 1 |
| | theta[6] | −0.641 | 0.476 | −1.557 | 0.23 | 14932 | 25140 | 1 |
| | theta[7] | −1.251 | 0.474 | −2.144 | −0.362 | 15336 | 26550 | 1 |
| | theta[8] | 1.615 | 0.566 | 0.548 | 2.677 | 19382 | 29906 | 1 |
| | theta[9] | 0.721 | 0.515 | −0.234 | 1.702 | 17127 | 28476 | 1 |
| | theta[10] | 1.615 | 0.566 | 0.568 | 2.693 | 19343 | 29784 | 1 |
| | theta[11] | 1.615 | 0.567 | 0.568 | 2.695 | 19363 | 27743 | 1 |
| | theta[12] | 0.166 | 0.492 | −0.766 | 1.078 | 15641 | 26534 | 1 |
| | theta[13] | 0.346 | 0.499 | −0.607 | 1.264 | 16040 | 27328 | 1 |
| numpyro CPU | | Estimation | | | | Diagnostics | | |
| | | mean | sd | hdi_3% | hdi_97% | ess_bulk | ess_tail | r_hat |
| | theta[0] | 1.615 | 0.562 | 0.613 | 2.722 | 18474 | 28342 | 1 |
| | theta[1] | 0.921 | 0.527 | −0.056 | 1.924 | 17348 | 27983 | 1 |
| | theta[2] | 1.612 | 0.565 | 0.545 | 2.667 | 19804 | 27694 | 1 |
| | theta[3] | 0.921 | 0.527 | −0.062 | 1.922 | 17238 | 27787 | 1 |
| | theta[4] | 1.141 | 0.539 | 0.122 | 2.139 | 17777 | 28747 | 1 |
| | theta[5] | 1.136 | 0.538 | 0.129 | 2.153 | 17642 | 27878 | 1 |
| | theta[6] | −0.643 | 0.473 | −1.51 | 0.256 | 15441 | 24593 | 1 |
| | theta[7] | −1.253 | 0.474 | −2.141 | −0.36 | 15186 | 24014 | 1 |
| | theta[8] | 1.614 | 0.567 | 0.555 | 2.678 | 19272 | 27497 | 1 |
| | theta[9] | 0.718 | 0.515 | −0.253 | 1.69 | 17100 | 26777 | 1 |
| | theta[10] | 1.611 | 0.566 | 0.519 | 2.648 | 19627 | 29707 | 1 |
| | theta[11] | 1.612 | 0.568 | 0.556 | 2.701 | 19678 | 28018 | 1 |
| | theta[12] | 0.163 | 0.491 | −0.752 | 1.093 | 15522 | 25928 | 1 |
| | theta[13] | 0.339 | 0.499 | −0.584 | 1.294 | 15960 | 25830 | 1 |
| numpyro GPU | | Estimation | | | | Diagnostics | | |
| | | mean | sd | hdi_3% | hdi_97% | ess_bulk | ess_tail | r_hat |
| | theta[0] | 1.618 | 0.561 | 0.581 | 2.689 | 18841 | 29675 | 1 |
| | theta[1] | 0.925 | 0.527 | −0.06 | 1.914 | 17399 | 26758 | 1 |
| | theta[2] | 1.615 | 0.568 | 0.564 | 2.697 | 19508 | 27450 | 1 |
| | theta[3] | 0.923 | 0.527 | −0.057 | 1.929 | 17223 | 27357 | 1 |
| | theta[4] | 1.143 | 0.537 | 0.151 | 2.166 | 17784 | 29717 | 1 |
| | theta[5] | 1.139 | 0.541 | 0.157 | 2.196 | 17700 | 28834 | 1 |
| | theta[6] | −0.641 | 0.474 | −1.524 | 0.247 | 15313 | 25672 | 1 |
| | theta[7] | −1.252 | 0.476 | −2.139 | −0.354 | 15033 | 26341 | 1 |
| | theta[8] | 1.617 | 0.567 | 0.551 | 2.672 | 19673 | 26825 | 1 |

**Table 4** (*continued*)

| | | | | | | | |
|---|---|---|---|---|---|---|---|
| theta[9] | 0.722 | 0.515 | −0.275 | 1.67 | 17220 | 26307 | 1 |
| theta[10] | 1.613 | 0.566 | 0.559 | 2.689 | 19741 | 28897 | 1 |
| theta[11] | 1.614 | 0.566 | 0.546 | 2.681 | 19503 | 27841 | 1 |
| theta[12] | 0.166 | 0.492 | −0.754 | 1.097 | 15657 | 26378 | 1 |
| theta[13] | 0.342 | 0.498 | −0.576 | 1.307 | 15644 | 25302 | 1 |

**Notes.**

Note: theta[i] of Table means $\theta_i$ of the equation of 1PL-IRT.

**Table 5** Estimation results of ability parameters of test takers in 2PL-IRT for BRAIN data.

| pystan | | Estimation | | | | Diagnostics | | |
|---|---|---|---|---|---|---|---|---|
| | | mean | sd | hdi_3% | hdi_97% | ess_bulk | ess_tail | r_hat |
| | theta[0] | 1.18 | 0.531 | 0.203 | 2.181 | 17825 | 28333 | 1 |
| | theta[1] | 0.902 | 0.496 | −0.008 | 1.843 | 15625 | 27710 | 1 |
| | theta[2] | 1.254 | 0.518 | 0.296 | 2.235 | 18845 | 29397 | 1 |
| | theta[3] | 0.704 | 0.538 | −0.3 | 1.707 | 14418 | 26881 | 1 |
| | theta[4] | 1.136 | 0.503 | 0.226 | 2.111 | 17740 | 28371 | 1 |
| | theta[5] | 1.04 | 0.52 | 0.096 | 2.035 | 17265 | 28069 | 1 |
| | theta[6] | −0.509 | 0.394 | −1.235 | 0.242 | 12631 | 24324 | 1 |
| | theta[7] | −1.27 | 0.433 | −2.075 | −0.446 | 15054 | 26335 | 1 |
| | theta[8] | 1.458 | 0.572 | 0.369 | 2.512 | 17362 | 28080 | 1 |
| | theta[9] | 0.94 | 0.531 | −0.024 | 1.965 | 15271 | 26223 | 1 |
| | theta[10] | 1.767 | 0.575 | 0.696 | 2.852 | 18638 | 29705 | 1 |
| | theta[11] | 1.365 | 0.509 | 0.411 | 2.317 | 18758 | 29696 | 1 |
| | theta[12] | 0.332 | 0.453 | −0.51 | 1.193 | 13643 | 24442 | 1 |
| | theta[13] | 0.324 | 0.465 | −0.542 | 1.209 | 14035 | 24973 | 1 |
| numpyro CPU | | Estimation | | | | Diagnostics | | |
| | | mean | sd | hdi_3% | hdi_97% | ess_bulk | ess_tail | r_hat |
| | theta[0] | 1.183 | 0.528 | 0.182 | 2.157 | 18389 | 29763 | 1 |
| | theta[1] | 0.898 | 0.488 | 0.009 | 1.837 | 16627 | 28545 | 1 |
| | theta[2] | 1.256 | 0.515 | 0.288 | 2.225 | 18436 | 28429 | 1 |
| | theta[3] | 0.704 | 0.535 | −0.291 | 1.711 | 14966 | 25748 | 1 |
| | theta[4] | 1.134 | 0.501 | 0.19 | 2.071 | 17305 | 28056 | 1 |
| | theta[5] | 1.044 | 0.518 | 0.089 | 2.031 | 16558 | 26045 | 1 |
| | theta[6] | −0.505 | 0.394 | −1.225 | 0.262 | 12487 | 23078 | 1 |
| | theta[7] | −1.265 | 0.432 | −2.078 | −0.465 | 15003 | 25697 | 1 |
| | theta[8] | 1.455 | 0.567 | 0.419 | 2.539 | 18819 | 28511 | 1 |
| | theta[9] | 0.943 | 0.53 | −0.035 | 1.942 | 16732 | 27505 | 1 |
| | theta[10] | 1.766 | 0.573 | 0.644 | 2.801 | 17756 | 27724 | 1 |
| | theta[11] | 1.365 | 0.506 | 0.426 | 2.326 | 19432 | 29606 | 1 |
| | theta[12] | 0.334 | 0.453 | −0.51 | 1.197 | 12867 | 24284 | 1 |
| | theta[13] | 0.327 | 0.462 | −0.523 | 1.204 | 13323 | 26250 | 1 |

**Table 5** (*continued*)

| numpyro GPU | Estimation | | | | Diagnostics | | |
|---|---|---|---|---|---|---|---|
| | mean | sd | hdi_3% | hdi_97% | ess_bulk | ess_tail | r_hat |
| theta[0] | 1.186 | 0.527 | 0.226 | 2.197 | 17302 | 20916 | 1 |
| theta[1] | 0.897 | 0.488 | 0.001 | 1.827 | 16923 | 26620 | 1 |
| theta[2] | 1.251 | 0.517 | 0.292 | 2.22 | 15879 | 16665 | 1 |
| theta[3] | 0.702 | 0.532 | −0.303 | 1.695 | 16956 | 26275 | 1 |
| theta[4] | 1.133 | 0.5 | 0.204 | 2.08 | 18464 | 28238 | 1 |
| theta[5] | 1.04 | 0.515 | 0.081 | 2.007 | 16628 | 28830 | 1 |
| theta[6] | −0.507 | 0.392 | −1.248 | 0.228 | 13277 | 24086 | 1 |
| theta[7] | −1.264 | 0.433 | −2.076 | −0.458 | 14157 | 19916 | 1 |
| theta[8] | 1.455 | 0.565 | 0.395 | 2.504 | 17398 | 23571 | 1 |
| theta[9] | 0.942 | 0.528 | −0.051 | 1.925 | 17510 | 22661 | 1 |
| theta[10] | 1.762 | 0.573 | 0.644 | 2.807 | 16888 | 24176 | 1 |
| theta[11] | 1.363 | 0.504 | 0.414 | 2.302 | 17175 | 12944 | 1 |
| theta[12] | 0.33 | 0.449 | −0.503 | 1.18 | 13347 | 25251 | 1 |
| theta[13] | 0.33 | 0.463 | −0.529 | 1.203 | 13968 | 25446 | 1 |

**Notes.**

Note: theta[i] of Table means $\theta_i$ of the equation of 2PL-IRT.

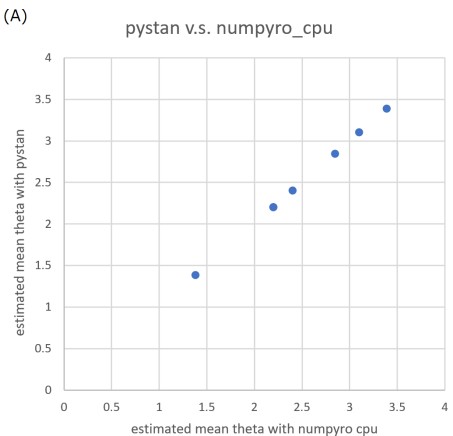
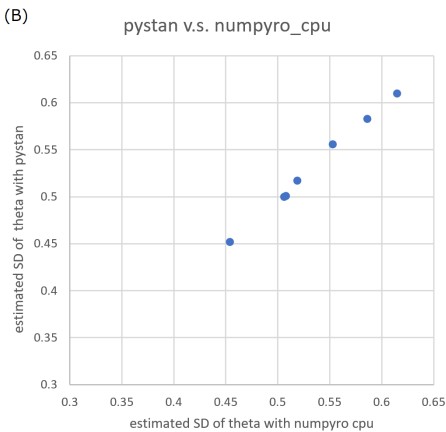

**Figure 1** **Representative scatter plots of estimated ability parameters of 1PL-IRT between numpyro and pystan for BONE data.** (A) Plot for mean of estimated ability parameters, (B) plot for SD of estimated ability parameters. Note: (A) and (B) are obtained from values from Table 2.

practitioners and researchers should select either pystan or numpyro based on the data size. Figure 11 shows our recommendation for selecting pystan and numpyro.

Tables 3 and 4 show that the total number of latent parameters may affect the usefulness of GPU for reducing the sampling time. In complex models, such as 2PL-IRT used in this study, numpyro GPU version may tend to be faster than numpyro CPU version. The effect of GPU on the sampling time should be evaluated in future studies.

This study had several limitations. First, we evaluated only Bayesian 1PL-IRT and 2PL-IRT. Further studies are needed to investigate the effectiveness of pystan and numpyro

(A)
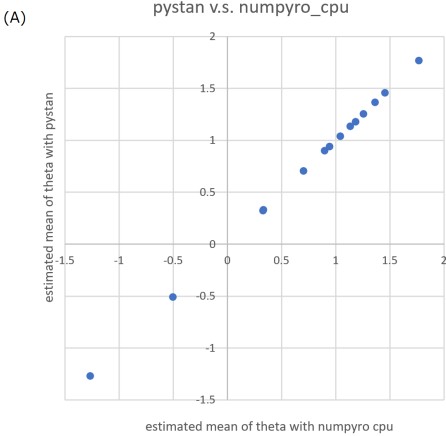

(B)
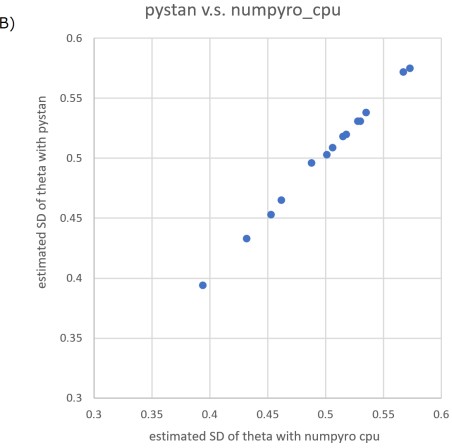

**Figure 2** **Representative scatter plots of estimated ability parameters of 2PL-IRT between numpyro and pystan for BRAIN data.** (A) Plot for mean of estimated ability parameters, (B) plot for SD of estimated ability parameters. Note: (A) and (B) are obtained from values from Table 5.

**Table 6** **Agreement of estimated ability parameters between pystan *vs.* numpyro CPU version, pystan *vs.* numpyro GPU version, and numpyro CPU version *vs.* numpyro GPU version.**

| Data type | IRT type | CCC between pystan *vs.* numpyro CPU version | CCC between pystan *vs.* numpyro GPU version | CCC between numpyro CPU version *vs.* numpyro GPU version |
|---|---|---|---|---|
| BONE data | 1PL-IRT | 1.000 | 1.000 | 1.000 |
| BONE data | 2PL-IRT | 1.000 | 1.000 | 1.000 |
| BRAIN data | 1PL-IRT | 1.000 | 1.000 | 1.000 |
| BRAIN data | 2PL-IRT | 1.000 | 1.000 | 1.000 |

**Notes.**
Note: The CCC values indicate almost perfect agreement. Because of the significant digits in the CCC calculation, "1.000" was used in Table 6. However, this "1.000" means that the CCC value was approximately 1. Actually, the CCC value of "1.000" was less than 1 (*e.g.*, 0.9999847).
Abbreviation: Lin's concordance correlation coefficients, CCC.

in other types of Bayesian models. Second, although we evaluated the sampling time, we used the simulation data instead of real-world data. Future studies should use real-world data to evaluate the sampling time. Third, we mainly used Google Colaboratory. Although Google Colaboratory has several merits (*e.g.*, ease of use and availability), our experiments were performed using limited types of hardware. Fourth, because the data and purpose of our study are different from those of *Nishio et al. (2020)*, it is impossible to compare our results with those from that article.

**Table 7  Agreement of estimated ability parameters between pystan *vs* numpyro_cpu, pystan *vs* numpyro_gpu, and numpyro_cpu *vs* numpyro_gpu in fewer samples.**

| Data type | IRT type | Number of samples per one chain | CCC between pystan and numpyro_cpu | CCC between pystan and numpyro_gpu | CCC between numpyro_cpu and numpyro_gpu |
|---|---|---|---|---|---|
| BONE data | 1PL-IRT | 10 | 0.986 | 0.994 | 0.995 |
| BONE data | 1PL-IRT | 20 | 0.986 | 0.998 | 0.992 |
| BONE data | 1PL-IRT | 40 | 0.992 | 0.999 | 0.994 |
| BONE data | 1PL-IRT | 80 | 0.997 | 0.999 | 0.998 |
| BONE data | 1PL-IRT | 160 | 0.999 | 1.000 | 0.999 |
| BONE data | 1PL-IRT | 320 | 1.000 | 1.000 | 1.000 |
| BONE data | 1PL-IRT | 640 | 1.000 | 1.000 | 1.000 |
| BONE data | 1PL-IRT | 1,280 | 1.000 | 1.000 | 1.000 |
| BONE data | 1PL-IRT | 2,560 | 1.000 | 1.000 | 1.000 |
| BONE data | 1PL-IRT | 5,120 | 1.000 | 1.000 | 1.000 |
| BONE data | 2PL-IRT | 10 | 0.965 | 0.971 | 0.998 |
| BONE data | 2PL-IRT | 20 | 0.987 | 0.990 | 0.998 |
| BONE data | 2PL-IRT | 40 | 0.995 | 0.994 | 0.999 |
| BONE data | 2PL-IRT | 80 | 0.990 | 0.992 | 0.999 |
| BONE data | 2PL-IRT | 160 | 0.998 | 0.998 | 0.999 |
| BONE data | 2PL-IRT | 320 | 1.000 | 1.000 | 1.000 |
| BONE data | 2PL-IRT | 640 | 0.999 | 1.000 | 1.000 |
| BONE data | 2PL-IRT | 1,280 | 1.000 | 1.000 | 1.000 |
| BONE data | 2PL-IRT | 2,560 | 1.000 | 1.000 | 1.000 |
| BONE data | 2PL-IRT | 5,120 | 1.000 | 1.000 | 1.000 |
| BRAIN data | 1PL-IRT | 10 | 0.983 | 0.985 | 0.999 |
| BRAIN data | 1PL-IRT | 20 | 0.994 | 0.990 | 0.999 |
| BRAIN data | 1PL-IRT | 40 | 0.999 | 0.999 | 0.999 |
| BRAIN data | 1PL-IRT | 80 | 1.000 | 0.999 | 1.000 |
| BRAIN data | 1PL-IRT | 160 | 1.000 | 1.000 | 1.000 |
| BRAIN data | 1PL-IRT | 320 | 1.000 | 1.000 | 1.000 |
| BRAIN data | 1PL-IRT | 640 | 1.000 | 1.000 | 1.000 |
| BRAIN data | 1PL-IRT | 1,280 | 1.000 | 1.000 | 1.000 |
| BRAIN data | 1PL-IRT | 2,560 | 1.000 | 1.000 | 1.000 |
| BRAIN data | 1PL-IRT | 5,120 | 1.000 | 1.000 | 1.000 |
| BRAIN data | 2PL-IRT | 10 | 0.990 | 0.996 | 0.997 |
| BRAIN data | 2PL-IRT | 20 | 0.997 | 0.997 | 0.999 |
| BRAIN data | 2PL-IRT | 40 | 0.999 | 0.999 | 1.000 |
| BRAIN data | 2PL-IRT | 80 | 0.999 | 0.999 | 1.000 |
| BRAIN data | 2PL-IRT | 160 | 1.000 | 1.000 | 1.000 |
| BRAIN data | 2PL-IRT | 320 | 1.000 | 1.000 | 1.000 |
| BRAIN data | 2PL-IRT | 640 | 1.000 | 1.000 | 1.000 |
| BRAIN data | 2PL-IRT | 1,280 | 1.000 | 1.000 | 1.000 |
| BRAIN data | 2PL-IRT | 2,560 | 1.000 | 1.000 | 1.000 |

**Table 7** (*continued*)

| Data type | IRT type | Number of samples per one chain | CCC between pystan and numpyro_cpu | CCC between pystan and numpyro_gpu | CCC between numpyro_cpu and numpyro_gpu |
|---|---|---|---|---|---|
| BRAIN data | 2PL-IRT | 5,120 | 1.000 | 1.000 | 1.000 |

**Notes.**

Note: (i) Except for the number of samples per one chain (num_samples), the parameters for sampling using Markov chain Monte Carlo method are the same as those of the main experiment. (ii) In the main experiment, number of samples per one chain is 8000. (iii) Estimated mean of ability parameter was used in calculating CCC. (iv) Rhat values are not always less than 1.10 in this experiment.

Abbreviation: Lin's concordance correlation coefficients, CCC.

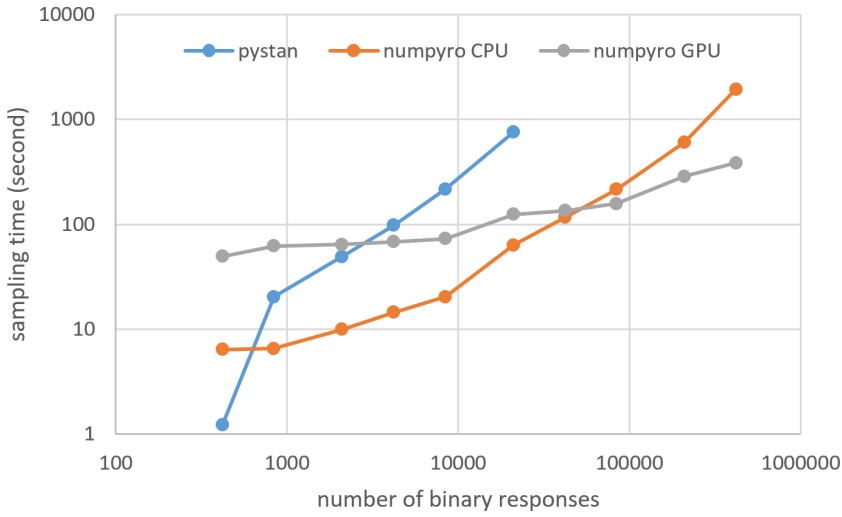

**Figure 3** Sampling time of 1PL-IRT for simulation BONE data.

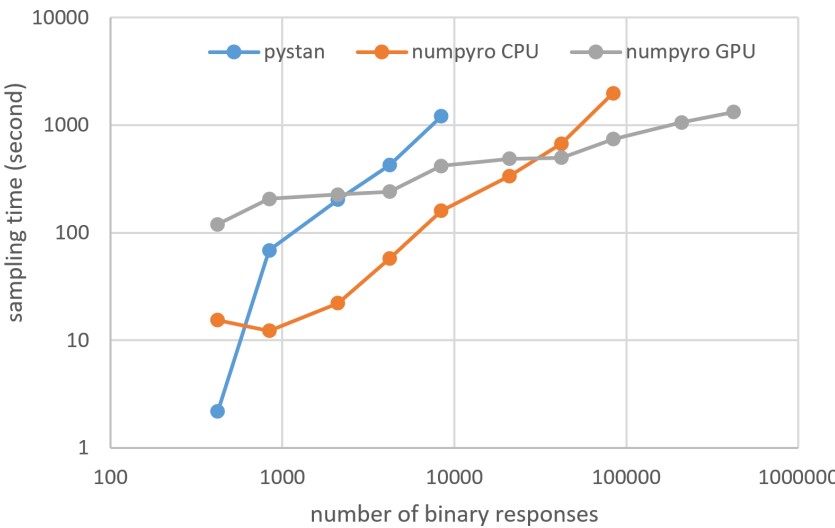

**Figure 4** Sampling time of 2PL-IRT for simulation BONE data.

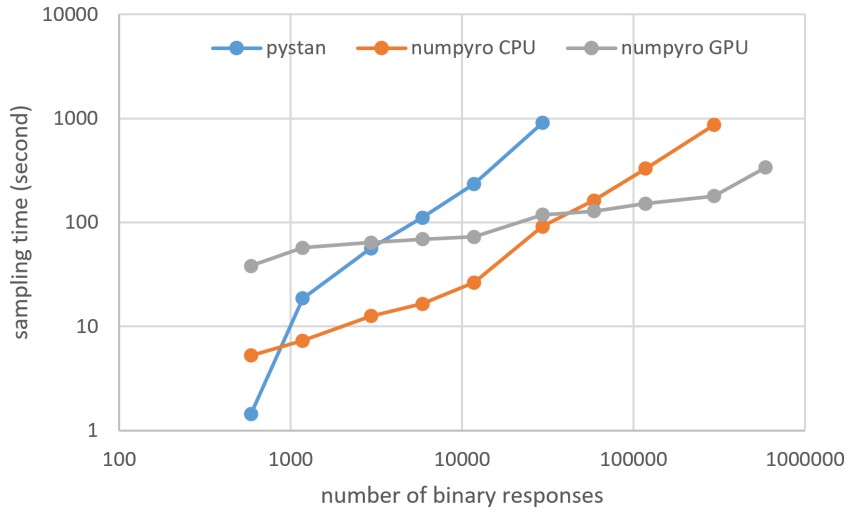

**Figure 5** Sampling time of 1PL-IRT for simulation BRAIN data.

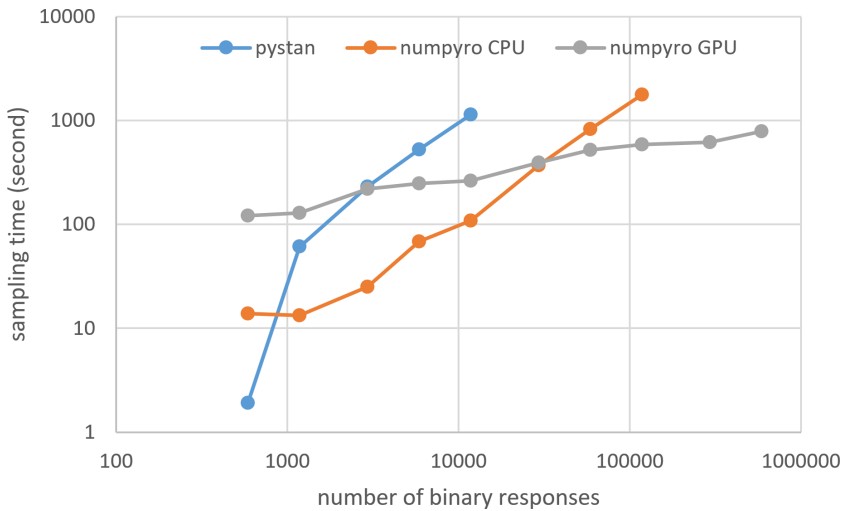

**Figure 6** Sampling time of 2PL-IRT for simulation BRAIN data.

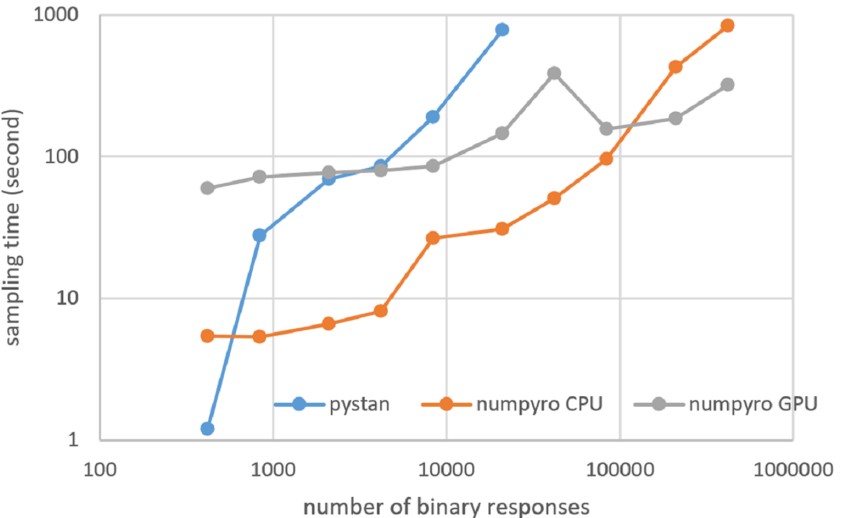

**Figure 7** Sampling time of 1PL-IRT for simulation BONE data on local workstation.

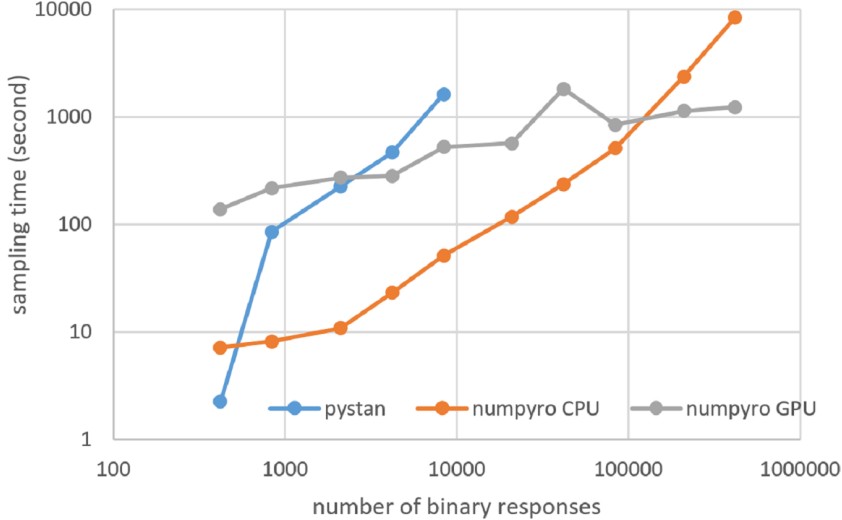

**Figure 8** Sampling time of 2PL-IRT for simulation BONE data on local workstation.

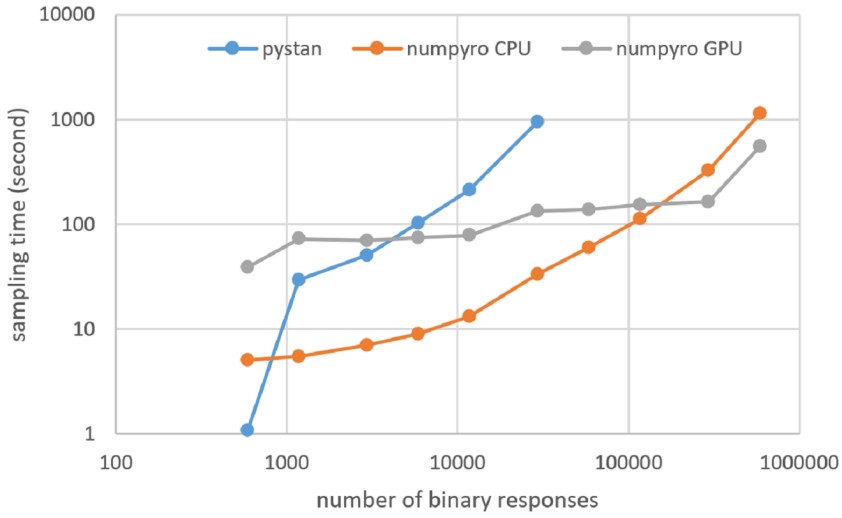

**Figure 9** **Sampling time of 1PL-IRT for simulation BRAIN data on local workstation.**

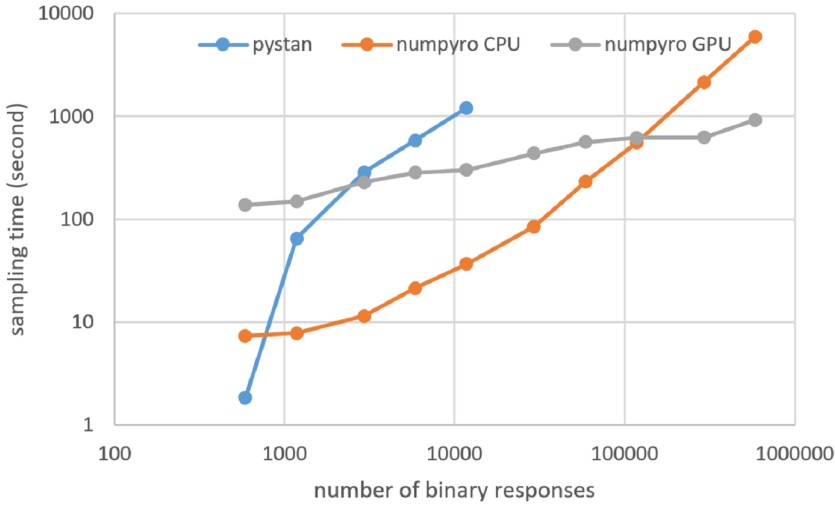

**Figure 10** **Sampling time of 2PL-IRT for simulation BRAIN data on local workstation.**

## CONCLUSIONS

The current study demonstrated that both pystan and numpyro were effective in the estimation for 1PL-IRT and 2PL-IRT of the BONE and BRAIN data. Our results show that the two libraries yielded similar estimation results. In addition, our results of sampling time show that the fastest libraries differed based on the total number of binary responses.

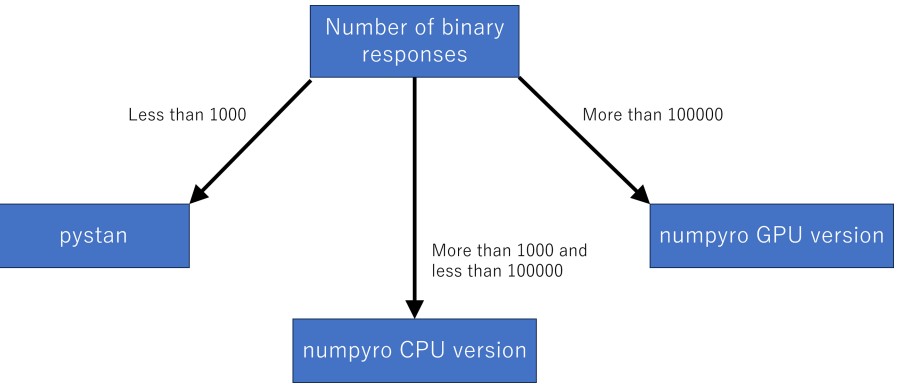

**Figure 11** **Recommendation for selecting pystan and numpyro.**

### Funding

This work was supported by JSPS KAKENHI (Grant Number: 22K07665 and 23K17229). There was no additional external funding received for this study. The funders had no role in study design, data collection and analysis, decision to publish, or preparation of the manuscript.

### Grant Disclosures

The following grant information was disclosed by the authors:
JSPS KAKENHI: 22K07665, 23K17229.

### Competing Interests

Eiji Ota is employed by Futaba Numerical Technologies. The other authors declare that they have no competing interests.

### Author Contributions

- Mizuho Nishio conceived and designed the experiments, performed the experiments, analyzed the data, performed the computation work, prepared figures and/or tables, authored or reviewed drafts of the article, and approved the final draft.
- Eiji Ota performed the experiments, performed the computation work, prepared figures and/or tables, authored or reviewed drafts of the article, and approved the final draft.
- Hidetoshi Matsuo analyzed the data, authored or reviewed drafts of the article, and approved the final draft.
- Takaaki Matsunaga analyzed the data, authored or reviewed drafts of the article, and approved the final draft.
- Aki Miyazaki analyzed the data, authored or reviewed drafts of the article, and approved the final draft.
- Takamichi Murakami conceived and designed the experiments, authored or reviewed drafts of the article, supervision, and approved the final draft.

## Data Availability

The code is available at GitHub and Zenodo:

- https://github.com/jurader/irt_pystan_numpyro

- jurader. (2023). jurader/irt_pystan_numpyro: First release (1st_release). Zenodo. https://doi.org/10.5281/zenodo.8187939

The data is available at: Nishio M, Akasaka T, Sakamoto R, Togashi K. Bayesian Statistical Model of Item Response Theory in Observer Studies of Radiologists. Acad Radiol. 2020 Mar;27(3):e45-e54. doi: 10.1016/j.acra.2019.04.014. Epub 2019 May 28.

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
