# Peer review of "Comparison between pystan and numpyro in Bayesian item response theory: evaluation of agreement of estimated latent parameters and sampling performance"

_PeerJ Computer Science, doi:10.7717/peerj-cs.1620_

## Round 0.1 · original submission · Major Revisions

· Academic Editor

Major Revisions

In my opinion this paper requires major revisions. The first reviewer's comments in terms of matching title and abstract to the actual contributions of the paper must be addressed. The actual contributions are not clear, to me or either of the reviewers, it is important to rewrite the abstract to make these clearer.
The points raised by reviewer 2 in regards to presentation and in particular, in regards to the results need to be addressed.

One of the main contributions seems to be the performance analysis, however, this is not entirely rigorous. Additional depth is required here as noted by both reviewers.

·

Basic reporting

The language is clear and easy to follow. References and background are given for the application side, but severely lacking regarding MCMC methods. Data and source code is published.

Experimental design

The article's title and purpose section make rather broad claims ("The purpose of this study is to compare two libraries dedicated to Markov chain Monte Carlo method"). The comparison undertaken is however restricted to a specific application and quite shallow in that it observes performance differences, but does not investigate what they are caused by. Beyond that, it seems like a straightforward application of existing Bayesian inference methods and software to a problem that is useful in applications, but not particularly challenging to solve. It does therefore not become clear how this work contributes new knowledge.

The fact that both numpyro and pystan eventually deliver near identical results is expected, since MCMC methods can be mathematically proven to converge to the desired posterior; any deviation would therefore be an implementation error or insufficient sampling. Both pystan and numpyro support various MCMC methods, which clearly affect solver efficiency. The paper does not indicate the MCMC methods used however, making further comparison questionable. Both packages may in fact use the same method by default; in this case, a meaningful performance investigation should look into how posterior evaluations are handled etc., instead of treating the packages as black boxes. A reasonable comparison in MCMC performance between two methods should be in terms of accuracy vs. computational cost, and not with a fixed number of samples. Further, the CPU/GPU comparison is biased towards GPU: The authors indicate they use only two CPU cores, but a high-end GPU. A more useful practical result might be comparing performance per hardware cost.

Validity of the findings

Results are presented well. Conclusions are however limited by the issues above.

·

Basic reporting

The authors evaluate two popular packages for Bayesian modeling regarding their performance for two IRT models, using two medical data sets for illustration. The authors report that pystan and numpyro estimate similar parameters for both one and two parameter logistical IRT models for both data sets. They also report sampling times for different sizes of simulated data sets from the posterior predictive distribution for inference purposes.

The major contributions I can currently see in this manuscript are:
-showing that these two IRT models can be fit with pystan and numpyro
-showing that both packages yield similar results
-providing an indication of which package yields these similar results faster, given a range of sample sizes drawn from the posterior predictive distribution

Some substantial limitations here lie in the constraints on the range of sample sizes estimated, as the authors mention themselves.

While general performance evaluations for these packages are available, more concrete information on how they perform for these types of IRT models has to my knowledge not been reported on so far.

That said, I see room for improvement of this manuscript in these major areas:
-clarity with which the value it provides is communicated
-generalizability of performance evaluations for researchers

I would suggest these specific changes to address them:

1) Run the same analyses on personal computers (desktop or laptop) and include the results.
The analysis scripts should not need too much adaptation here while the computation time might become somewhat bothersome for the larger sample sizes, but by benchmarking the analyses using local machines the value to readers would be increased, as not every researcher has access or is permitted (e.g. because of privacy concerns) to use cloud computing services. This is mostly a pragmatic concern, but may be quite important to potential readers.

2) Expand on for what purposes the IRT models can be used.
This is somewhat redundant to large parts of the literature and a brief summary with reference to other discussions would probably suffice, but for a potential reader of the manuscript it would be useful to have at least a rough idea about what they can do with these tools and a starting point where to look for more information. One or two sentences in the introduction each for the more common psychometric use cases as well as the use cases in a medical context, respectively, would probably be sufficient here.

3) Compare the results of your inference to those of your original paper of which you used the data, as far as this is possible. Also compare the performance metrics, which may have to be done in conjunction with the first suggestion to keep the metrics comparable.

4) Create a table for the central results. I don't mean the results of the parameter estimation but the results that are central to your study: Indices of agreement between the packages, between the packages and the results of the previous paper in which these data were used as well as runtimes for the various permutations of settings. Some of this is already present in graphs and the additional information would lie in the numbers written down instead of plotted. Additionally, recommendations for researchers (which you already discuss) regarding strengths and weaknesses of these tools lend themselves well to being summarized in a graphical format, either a table or a flow chart or something along those lines.

Overall these suggestions aim at clarifying to potential readers what they can learn for their own research by reading this manuscript. What can they expect in terms of accuracy issues and computational costs if they decide to use these tools themselves? As a side effect, this would also serve to further distinguish the present results from those you reported on the same data set in the 2022 paper.

On a more minor note, in Figure 2, panel A, a note saying 'plot area' is left in the plot area. While this is an accurate label, I assume it was left there on accident and should probably be removed.

Experimental design

no comment

Validity of the findings

See the comments regarding basic reporting

Additional comments

no comment

---

## Round 0.2 · Minor Revisions

· Academic Editor

Minor Revisions

Of the points raised by reviewer 2 point 2 remains to be addressed.

·

Basic reporting

no comment

Experimental design

no comment

Validity of the findings

no comment

Additional comments

The authors adressed my previous comments appropriately and as such I only have some very minor comments:

1:
Optional: I appreciate that the authors re-ran the experiments on their workstations and reported the results. I personally would put these in the main part of the manuscript rather than the appendix, as they generalize somewhat more easily than the results based on using a specific company's service (with the caveat that both are from a moment in time - hardware changes everywhere all the time), but that is more my personal preference and probably informed by my own research priorities.

2:
Table 6: "The CCC values indicate almost perfect agreement"

The values in the table have been rounded to 1, but given that 1 is the very highest value this coefficient can (!) take, distinguishing between >.999 and 1 matters here. A "merely" very high value is to be expected, while a maximal value can be indicative of technical issues in such contexts. As you asy: The values are *not* in perfect agreement, so this should be reflected in the reported values as well, which you already do in the text.

3:
I find the summary of the results in Figure 7 very helpful, thank you!

---

## Round 0.3 · accepted · Accept

· Academic Editor

Accept

As these were second, minor revisions I have not asked for another review of the revisions. All reviewer comments have now been addressed.